# Emergency dispatchers as instructors of laypersons in unplanned out-of-hospital deliveries - Interdisciplinary qualitative study

Jussi Hänninen[1]*, Hanna Toiviainen[2], Hilla Nordquist[1]

1 Department of Healthcare and Emergency Care, South-Eastern Finland University of Applied Sciences, Kotka, Finland, 2 Faculty of Education and Culture, Tampere University, Tampere, Finland

* jussi.hanninen@xamk.fi

## Abstract

### Introduction

Unplanned out-of-hospital delivery tasks are continuously increasing and are challenging to the emergency dispatcher, but there is scant research on them, especially from the perspective of instructing the layperson in delivery. In the Finnish model of one emergency response centre authority, the expertise of the emergency dispatcher is highlighted. However, it's necessary to develop collaborative and transformative expertise and cooperation in the maternity care service system to respond to risky and unexpected childbirth events. This study adopts concepts from cultural-historical activity theory in the framework of object-oriented activity systems and negotiated knotworking. Our aim is to describe and analyse the perspectives of emergency dispatchers and laypersons when the emergency dispatcher instructs the layperson in handling an unplanned out-of-hospital delivery.

### Methods

This is an interdisciplinary qualitative study. The study data consists of stories written by emergency dispatchers (n = 31) and semi-structured interviews with laypersons (n = 5). The data was collected using both the method of empathy-based stories and semi-structured interviews. The data was analysed using qualitative theory-driven content analysis, data-based theory-driven thematic analysis and text mining.

### Results

The emergency dispatchers' and laypersons' descriptions of their actions in the examined delivery situations were structured as elements of activity systems. The thematic analysis produced two themes containing discursive characteristics of negotiated knotworking, which were (i) script innovations requiring midwifery competence and (ii) emotion work.

**Data availability statement:** Data Availability statement Ethics approval This study received a favorable opinion (Statement 127/2023, attachment) from the Ethics Committee of the Tampere Region during the advance evaluation on September 5, 2023. The ethical approval for this research did not include the public publication of the data. Therefore, making the data publicly available would breach compliance with the protocol approved by the research ethics board (PLOS ONE Data Availability Policy). It is not possible to make the data publicly available for ethical reasons, as it contains sensitive, health-related information that could compromise the participants' privacy (ALLEA 2023; GDPR; TENK 2019). Health-related information (e.g., childbirth) constitutes special categories of personal data and must be handled with particular care, as it requires specific protection (GDPR). Other researchers can find more information about the Ethics Committee of the Tampere Region here: https://www.tuni.fi/en/research/respon-sible-science-and-research/research-integrity/ethics-committee-of-the-tampere-region and they can request access to the research data (available up-on request) via email: Secretary: Senior specialist Heikki Eilo heikki.eilo@tuni.fi or researchdata@tuni.fi Research autho-rization The Emergency Response Centre Administration granted the research permit regarding the method of empathy-based stories (MEBS) on December 8, 2023 (HAK-2316452, attachment). The issued research permit did not include permission for the public publication of the research data. Although the MEBS data has been pseudonymized, a risk still exists of identifying individual participants or of sensitive, confidential health-related informa-tion being accessed by unauthorized parties. The informed consent (ALLEA 2023; TENK 2019) provided by the emergency dispatchers for participation in this study did not include permission for the public publication of the research data, because they also refer to third parties (lay assistants and birthing persons). Health-related information (e.g., childbirth) constitutes a special category of personal data and must be handled with particular care, as it requires specific protection (GDPR). Other researchers can request access to the data from the Emergency Response Centre Agency https://112.fi/en/frontpage via email to: Head

## Discussion

The object of the emergency dispatcher's actions was the physical wellbeing of the person giving birth and the newborn, while the layperson's object was the childbirth experience, including the aforementioned and shared with the person giving birth. The formal script of childbirth services does not serve negotiated knotworking. Further, a midwife's participation in an emergency call, including video consultation, is desired. The emergency dispatcher should respect the object-oriented conscious agency of the layperson, providing ad hoc information for childbirth.

## Introduction

In Finland, the Emergency Response Centre Agency handles approximately two thousand tasks related to childbirth each year, and about one in ten of those defined as urgent ends with an unplanned childbirth event [1,2]. These childbirths are unexpected and progress quickly, with people operating under great pressure without routine [3,4]. Unplanned out-of-hospital deliveries without the presence of a health care professional (henceforth, OHD) are relatively rare in countries with developed health care, but their number is likely to increase as childbirth services are centralised [4–7]. Up to one-third of OHDs take place without medical assistance [8]. The emergency dispatcher (henceforth, ED) is often left with the responsibility of instructing the layperson (henceforth, LP) in performing the delivery. It is vital to pay attention to the development of collaborative and transformative expertise [9] as well as cooperation in the maternity care service system [see 10,11].

Finland uses the model of one emergency response centre authority, a multi-authority emergency response centre system. For example, an emergency call related to an OHD is not specifically transferred to a maternity hospital midwife, high-lighting the ED's expertise. The ED instructs and guides the caller, e.g., the LP who has to handle the birth, on how to act before help arrives. The actions of the ED are regulated by risk assessment guidelines, which do not include advice from a health care professional [12–15].

There is no ED expertise study related to OHDs. LPs' experiences of handling OHDs under the direction of EDs have previously only been examined in one study on Norwegian fathers [13], which emphasised the importance of the support and instruction provided by a health care professional in a stressful situation. The aim of this study is to describe and analyse the perspectives of EDs and LPs on handling unexpected OHDs. The perspectives are conceptualised in the cultural-historical activity theory (CHAT) framework as object-oriented activity [16]. The framework provides a point of departure for analysing collaborative and transformative expertise. Attention is focused on the meanings given to the activity by different parties, in this case, professionals and LPs, and possibly on constructing the shared object and meaning of the activity. Cooperation requiring rapid solutions has been described as negotiated knotworking, which includes surprises and typically some deviation from the script and improvisation [9].

of Project Management Office: Mrs. Ullamaija Nenonen ullamaija.nenonen@112.fi or pro-jektitoimisto@112.fi The informed consent (ALLEA 2023; TENK 2019) provided by the lay assistants for participation in this study did not include permission for the public publication of the research data. Although the thematic interview data have been pseudonymized, there remains a risk of identifying individual partici- pants or of sensitive, confidential health-related information being accessed by unauthorized parties. When obtaining informed consent from the lay assistants, permission for public data sharing was not requested, as this would not comply with the European GDPR, to which we must adhere. Opening the data retrospectively without informing the participants would not align with the good scientific practice as defined by the Finnish National Board on Research Integrity (TENK). First authorship agreement As the corresponding author and custodian of the data, I am committed to doing everything possible to ensure that other researchers are granted access to both MEBS and thematic interview data (available upon request) when necessary. According to the research ethics and data policy of Tampere University, the research data can only be held by the researcher if they are not employed by the university. Since I am not in an employment relationship with Tampere University, I am the data controller for the research, not the university. Personally, as a researcher, I support the PLOS ONE Data Availability Policy and the mission of Open Science. However, I am bound by ethical guidelines (ALLEA 2023; TENK 2019) and legislation (GDPR). The principle 'as open as possible, as closed as necessary' applies in this context (PLOS ONE Data Availability policy). email: jussi.hanninen@xamk.fi References European Federation of Academies of Sciences and Humanities ALLEA (2023). https://allea.org/code-of-conduct/ Finnish National Board on Research Integrity TENK (2019). The Ethical Principles of Research with Human Participants and Ethical Review in the Human Sciences in Finland. Finnish National Board on Research Integrity TENK Guidelines 2019. https://tenk.fi/sites/default/files/2021-01/Ethical_review_in_human_sciences_2020.pdf General Data Protection Regulation GDPR 9§. https://eur-lex.europa.eu/legal-content/FI/TXT/?uri=uriserv:OJ.L_.2016.119.01.0001.01.ENG&toc=OJ:L:2016:119:TOC PLOS ONE Data Availability Policy. https://journals.plos.org/plosone/s/data-availability.

The research questions are:

1. How are the dynamics and significance of activity in an OHD constructed from the perspectives of EDs and LPs?

2. What kinds of characteristics and obstacles of negotiated knotworking can be observed in the activity of EDs and LPs during an OHD?

## Materials and methods

This is an interdisciplinary qualitative study utilising text mining. The target group consists of emergency response centre dispatchers (EDs) working in Finland and laypersons (LPs) who, under the dispatchers' guidance, handled unplanned out-of-hospital deliveries (OHDs).

### Ethics approval and consent to participate

The study received a favourable opinion (Statement 127/2023) in the advance evalu- ation of the Human Sciences Ethics Committee of the Tampere region on 5 Septem- ber 2023.

In accordance with good research practice [17], all informants were given the essential information about the study, the voluntary nature of participation, protection of anonymity and confidentiality in advance. The first author's doctoral researcher role, midwife profession, motives and relationship with the phenomenon being stud- ied were explained to the subjects in advance. Informed consent was requested in writing and confirmed orally at the beginning of the interview.

### Theoretical background

**Expertise as an object-oriented and contradiction-driven activity.** The activity of the parties is analysed as activity systems consisting of both object-oriented individual actions and the broader communal and societal boundary conditions of the actions. For example, the person giving birth is an active subject in their activity system, using material tools and symbols, so-called mediating tools, which are characteristic of childbirth and have developed in the culture. Childbirth activity is directed at and motivated by the object, a positive overall childbirth experience, in which case the empowerment of the person giving birth emerges as the outcome. Both formal and unspoken rules regulate childbirth activity, as well as tacit norms that apply to births. Community is a central element comprised of people who share the object, in this case, assist or are otherwise socially involved in the birth. Division of labour defines the tasks and responsibilities of activity, assigning special roles for the subject of childbirth and the participating professionals and laypersons [cf. 16,11].

For its part, the community of the person giving birth determines who can be sub- jects in the childbirth activity, and it often consists of a childbirth support person import- ant to the person giving birth. An OHD, where a health care professional cannot be present, changes the structure of the activity system, the division of labour and the role of the support person from a community background support to the subject, the LP [9].

**Funding:** Financial Disclosure Statement / Funding Publication and Production Services of the South-Eastern Finland University of Applied Sciences covers the Article Processing Charge

**Competing interests:** Competing interests The authors declare that they have no competing interests.

**Abbreviations:** CHAT, cultural-historical activity theory; ED, emergency dispatcher; LP, layperson; MEBS, method of empathy-based stories; OHD, unplanned out-of-hospital delivery (without the presence of a health care professional); UWSED, unwell-script; WSED, well-script.

In these unexpected and risky childbirth events, the LP may become confused about their role and worry about the wellbeing of the person giving birth and of the baby, and hide their negative feelings to protect the person giving birth [13,18–20]. This study focuses on the LP's activity and cooperation with the ED [see 21], i.e., the relationship between the activity systems of the two subjects in an unplanned childbirth event.

Typically, after calling the midwife of the delivery room, the LP in need of help almost without exception calls the emergency telephone number as they become aware that they will not make it to the maternity hospital in time [11,13,22]. Theoretically, the emergency call is a mediating tool in the communication between the ED and the LP. In Finland, the progress of the discussion between ED and LP during the emergency call is guided by an established plan, i.e., the national risk assessment instructions on processing health care tasks at the Emergency Response Centre [see 23]. This mediating artifact regulating the course of interaction is a progression chart, which in activity theoretical research has been referred to as the formal script of cooperation. Disturbances and innovations observed in the practical work can be interpreted as deviations from the script and as expressions of developmental contradictions in activity systems [24]. Fig 1 summarises the theoretical background examined above, the construction of expertise as object-oriented activity in the disruption-prone interactional relationships between activity systems [9].

**Applying negotiated knotworking.** In health care service systems, permanent teams are replaced with flexible expert configurations as non-linear cooperation between employees or organisations occurs in temporary and dynamic networks, knots, formed around a specific challenging task [25]. Engeström [26] has suggested 'negotiated knotworking' as "*an emerging mode of activity describing the fluent and constantly changing settings of collaboration in client-oriented work.*" Unlike traditional networking, negotiated knotworking is an improvised form of collaboration where the activity is not based on strict rules or hierarchies. Negotiated knotworking manifests as the subjects' mutual design or problem-solving effort in activity systems oriented towards a shared object. Border crossings between different professions and competence areas and destabilisation of established power structures can be observed in the work [26].

According to Engeström [9], negotiated knotworking, like collective activity systems, is the 'spearhead' of research when the research interest is the development of collaborative and transformative expertise. Engeström and Pyörälä [27] argue that negotiated knotworking is one of the core elements in the development of medical expertise. We suggest that negotiated knotworking is an important consideration in the analysis of OHDs, in which unexpected and risky events pose challenges to expertise and require an instant response and negotiation of temporary practices [25,26]. In this study, the context of the setting depicted is the collaboration of the ED and the LP and negotiated knotworking to achieve a positive overall childbirth experience that includes childbirth safety. Negotiated knotworking has been analysed in multiple health and social care work studies. Recent examples include studies on opportunities for computer-aided knotworking for health care agents [28], and on forms of multi-professional cooperation and learning in health care [29].

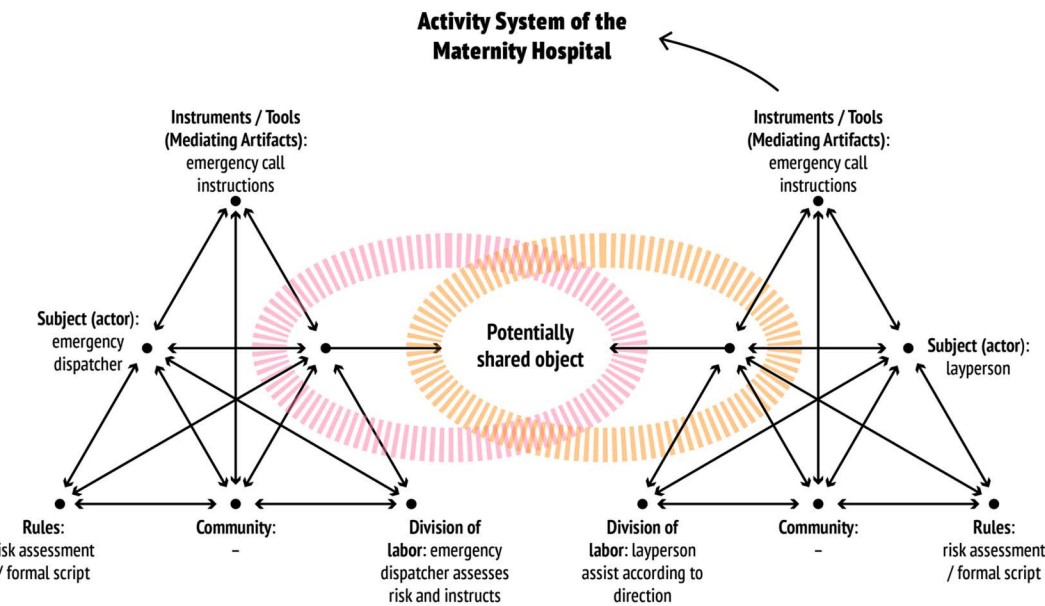

**Fig 1. Construction of emergency dispatcher's expertise in the interaction between object-oriented and contradiction-driven activity systems.**

Engeström et al. [25] discerned three dimensions of knotworking: i) socio-spatial, ii) temporal, and iii) ethical. Each of these dimensions is discursively elaborated on in the participants' talk. Thus, the socio-spatial dimension is manifested in the questions of who is and who should be involved in the collaborative activity at hand. The temporal dimension is addressed in the informants' accounts of the pulsating activity, in which knots are tied and untied in various rhythms of time. In health care activities, the ethical dimension is a central theme reflected in questions of how to meet vulnerability, how responsibility and power are renegotiated among the participants in acute situations. In addition, collaborating parties may assess the tools and practises that mediate the knotworking actions in iv) the material–instrumental dimension. Designing and implementing tools may stabilise and institutionalise instant knotworking practices [25].

**Sampling and recruitment.** The material used in this study is the stories (n = 31) collected from Finnish EDs employed by the Emergency Response Centre Administration in Spring 2024 using the method of empathy-based stories (MEBS) [30]. The EDs were recruited between January 30 and April 1, 2024. This data was complemented by semi-structured interviews (n = 5) of LPs who had handled an OHD under the guidance of an ED between 2019 and 2024 [31]. The LPs were recruited between December 1, 2023, and March 29, 2024.

The Emergency Response Centre Administration granted the research permit regarding MEBS in December 8, 2023. The invitation to participate in the study was sent to all EDs in Finland (N > 400). The contact person named by the Emergency Response Centre Administration forwarded the invitation and the link to the electronic questionnaire to EDS. The invitation was extended four times.

To obtain the semi-structured interview material, the few LPs in Finland who have performed an OHD under the guidance of an ED were approached through the people who had given birth and were interviewed in a previous study on childbirth experiences [11]. An interview invitation was also published on the Finnish Midwifery Society's social media, and press releases were published in the Finnish media. Each recruitment strategy required the consent of the person giving birth for the layperson to participate in the study. Communication took place by telephone, letter and email.

**Data collection.** The empathy-based stories (MEBS) method sought the perspectives of EDs, and the semi-structured interview material sought the perspectives of LPs. The first author was sensitive to the ability of both informant groups

to participate in the study and give their answers, leading to two different but still comparable data types. The first author designed the research, collected the data, and prepared the data transcriptions for the analysis.

The MEBS data was collected by the first author between 6th February and 6th April 2024. In addition to writing the story, background information was also collected from EDs in order to describe the material: level of education, occupational group, age, gender, work experience as an emergency dispatcher and the number of deliveries performed by instructed laypersons. The EDs empathised with the fictitious situation indicated in the frame story, i.e., the script (data in S1 Appendix), and based on that, wrote a story on the Webropol platform. The variable factor was the successful (well-script, WSED) or unsuccessful (unwell-script, UWSED) completion of the OHD from the perspective of a professional. The response time was not limited in advance, but a deadline of two months was set for the response. The investigator could not verify where the stories were written or whether other persons were present. The survey and reporting tool randomly selected which version of the frame story each informant received. The scope of the stories written based on the first frame story (WSED) (n = 15) was 2202 words, and the average length was 147 words. The scope of stories written based on the second framework report (UWSED) (n = 16) was 1859 words, and the average length was 116 words.

To collect the semi-structured interview material, Zoom interviews were conducted between 14th December 2023 and 14th March 2024. The themes were based on a theoretical preliminary understanding but left room for differing interpretations and views [32]. The interview themes (data in S2 Appendix) were derived from a literature review [e.g., 13,18–20], the researcher's expertise and the concepts of CHAT [33]. The first author was the interviewer. The framework of the semi-structured interview was tested on a voluntary LP belonging to the target group. When the pilot interview proved successful, it was included as part of the material of this study. No material or interview themes were sent to the interviewees in advance. The subjects participated in the Zoom interview from their homes and no outsiders were present during the interviews. The LP interviews were recorded. The total length was 5 hours and 40 minutes (min 69 and max 94 minutes per interview, 85 minutes on average). The material was transcribed (total of 48,837 words, an average of 12,210 words in Finnish). The transcripts were not submitted to informants for comment. However, two subjects wanted to supplement their interview by email, and the supplements were included in the material of this study.

## Analysis

To answer the first research question, qualitative theory-driven activity system content analysis was used. However, the processing of the data did not aim at abstraction, typical of content analysis [see 34], but rather at a precise description of the elements of the activity system [see 16] and the disturbances experienced in them, based on the expressed and manifest contents. The elements of both EDs' and LPs' activity systems were analysed from the data. Parts of the MEBS and semi-structured interview data related to the elements of the activity systems were identified and isolated from the data for further analysis and interpretation. The analysis first focused on the way in which each perspective (ED and LP) defines and structures the elements of the activity. Then, disturbances associated with the elements by the respondents were examined, characteristic of which were various expressions of problems and obstacles encountered in the work. Finally, the structures and disturbances constructed by the elements of the activity systems were written out as the result text.

To answer the second research question, respondents' (ED and LP) descriptions of cooperation in the unexpected delivery situation were separated from the data and interpreted as expressions of negotiated knotworking. The analysis method used was data-based theory-driven thematic analysis [35], which can be referred to as abductive or, as Xu and Zammit [36] do, as a hybrid approach where *"insiders' insights were interpreted through the theoretical lens while also allowing the participants to express themselves"*. Initially, discursive manifestations of negotiated knotworking were extracted from research literature and used to identify preliminary data points and potential themes from the responses in the MEBS and interview data. Specifying the final themes required careful data-based close reading and interpretation. Theory-based terms alone were insufficient to capture the specific features of the studied cooperation, including the unpredictability and sometimes chaotic nature of the unexpected childbirth situation. The themes formed were interpreted in cooperation between the research group and set proportionately to the activity systems. Representative data excerpts are shown to confirm the interpretations made.

To examine the impact of the variation [30], the analysis of the MEBS data was continued using a frame-story-specific (WSED, n = 16/ UWSED, n = 15) comparison. The basis for further interpretations regarding both research questions was strengthened using distant reading, which was carried out by text mining as an external expert service using the tm [37] and quanteda [38] packages of the statistics program R. The results were reported by interpreting the meanings of word frequencies, which specified the benefits of distant reading produced by the visualisation of word frequencies.

## Results

The EDs (n = 31) that responded to the study were mainly women aged 31–35. Their average work experience was 7.5 years. Their typical basic education was vocational upper secondary education, while some had a bachelor's degree. Almost all had completed the Emergency Response Centre Operator degree. One-third had a health care degree, and none was a midwife. During their emergency dispatcher career, approximately 80 per cent had instructed one or more OHDs handled by a layperson. The LPs (n = 5) were a mother, a mother-in-law, a partner and two voluntary labour support persons belonging to the community of the person giving birth. Each of them had become a layperson assistant unasked and accidentally. Each of them had previous experience of a hospital delivery, but none of them had assisted in a delivery.

### Dynamics and meaning of an unplanned delivery

**Subject: emergency dispatcher. Object:** In the EDs' stories (WSED and UWSED combined, n = 31), the most frequent words were "mother" and "child", describing the importance of the physical wellbeing of the two typical objects of activity in a childbirth task. In unexpected, risky and rapidly progressing unplanned childbirth events, decisions had to be made quickly as there was often not enough time for negotiation and construction of a shared object. The experiential nature of the activity was taken into account in some stories:

*The mother may find it difficult to concentrate on giving birth, and the childbirth experience may become poor; for example, tolerating pain is more difficult when the situation is unsafe and unplanned* (Participant 18, UWSED).

**Tools:** In their stories, the EDs gave the LPs relevant instructions by telephone to enable the delivery activity. In particular, the WSED stories (n = 15) highlighted the instructions' clarity, adequacy and repetition.

At the same time, in the EDs' stories, the relevant instructions describing both the tools and the object of the activity were primarily related to the newborn's wellbeing, such as stimulation by drying, taking care of thermoregulation or starting basic resuscitation. The instructions related to the place and position of childbirth varied as long as the person giving birth was moved away from the toilet seat. Relevant instructions related to handling the delivery included, for example, uncovering the lower body and observing the presenting part in the birth canal, instructing the person giving birth to push during a contraction, and if the mother experiences severe bleeding, she is placed into the recovery position.

In particular, the UWSED stories (n = 16) expressed concerns about a situation in which both the mother and the baby were unwell, and the ED had to guide the LP in treating two critical patients requiring immediate action, forced to prioritise:

*An unconscious, bleeding mother should be placed on her side, and resuscitation should be started for the baby. One layperson cannot take care of both. The layperson is trying to do their best, as am I. I'm trying to give instructions and keep everyone alive. The low number of maternity hospitals has driven us to this point, where an emergency dispatcher will face situations like this.* (Participant 30, UWSED).

**Rules:** In the EDs' stories, the delivery activity was guided by national risk assessment instructions on handling health care tasks at the Emergency Response Centre, the formal script of cooperation. Deviating from it caused noticeable disturbances in the childbirth task.

Especially in UWSED stories, disturbances had typically been caused by the reluctance of the LP to view the external female genital organs, in which case the LP had questioned the ED's instruction to examine the progress of labour visually:

*Cultural differences affect, for example, whether a father or other male family member or loved one can even view what is happening in the private parts. This leads to hazardous situations if it is not actually known whether something abnormal is present.* (Participant 22, UWSED).

In the stories, the LP had sometimes feared that they would worsen the situation and, contrary to the ED's instructions, had not dared, for example, to loosen the umbilical cord around the baby's neck in fear of tearing it. This had led to a stillborn baby. These deviations from the script by the LP being instructed caused confusion in the EDs in the absence of risk assessment instructions or visual contact:

*I can only give a certain amount of support and instructions orally. Not everything can be handled remotely.* (Participant 5, WSED).

Disturbances have sometimes been caused by the lack of a tool, common language, between ED and LP, making it impossible to ask questions, give instructions, or understand them.

In UWSED stories, disturbances had been caused by a Finnish-born person giving birth who had been thoroughly prepared for a hospital birth. The disappointment about the unplanned childbirth event had been expressed as acting up towards the ED and concretely as a refusal to follow the rules, instructions given by the ED through the LP.

Many UWSED stories showed that directing the LP in a car on the way to the hospital particularly challenged the script's rules. Traffic noise interfered with communication, and often, the LP was driving, which diverted attention from receiving and following instructions and assisting the person giving birth. After the ED had instructed the LP to stop, difficult conditions in the front seat caused more disturbances as space was limited, and the seat was difficult to move.

**Subject: layperson. Object:** The object of activity of all LPs interviewed was the joint effort of the LP and the person giving birth to complete the labour safely in order to achieve a positive overall outcome. Only one LP experience became partially negative and was described as a survival story containing self-accusation. The other LPs achieved a positive overall childbirth experience as the person giving birth and the newborn were well:

*It was a natural, beautiful, good, and even sacred experience. Unique, unprecedented and incredible, so absolutely nothing negative. Pure plus.* (Participant 2, LP).

The LPs identified the physical wellbeing of the mother and the child as the primary object of the EDs' activity. They shared the experience of a narrow time window for negotiating a shared object.

**Division of labour:** Achieving a positive overall childbirth experience required the acceptance of the LP in the "childbirth bubble" of the person giving birth, either partially or completely. From the perspective of the division of labour in the activity system, this can be interpreted as a shared responsibility between the LP and the person giving birth while the ED provides meaningful instructions in the background:

*The emergency dispatcher didn't bring themselves to the focus but listened and followed the situation as necessary, giving support and instructions, but was not there inside our bubble. I didn't think there were three of us in this childbirth, but that there were just the two of us living this thing* (Participant 1, LP).

**Outcome:** The shared intimate experience – which the LP had rarely talked about afterwards with people other than their loved ones – often resulted in the strengthening of the relationship between the LP and the person giving birth:

*We have been on the same side of the table and looked in the same direction, experienced a shared positive trauma, a successful unplanned delivery outside of a hospital. Nothing could unite two people and a relationship more.* (Participant 3, LP).

Sometimes, the experience resulted in a special bond between the LP and the child born unplanned outside of a hospital.

**Tools:** Some of the LPs viewed themselves as subjects in their activity system. To handle the OHD, they needed relevant instructions from the ED and ad hoc information for delivery. The LPs appreciated detailed, concrete, and easy-to-understand instructions. Some felt that instruction from different EDs was not consistent.

The instruction related to the wellbeing of the newborn, to take care of the thermoregulation, was highlighted as relevant in many LPs' interviews. The instruction to move the person giving birth away from the toilet seat was viewed as relevant in many LPs' interviews. However, the instructions related to delivery, not to pull the baby, were often found confusing. In many LPs' interviews, the instructions to remove the foetal membranes from around the baby's head and on the assessment of the amount of lochia were highlighted as relevant.

The interviewed LPs also brought up certain situations related to handling unplanned childbirth events where they felt they had not received related instructions from the ED. Especially the lack of instructions related to considering the intimacy of the person giving birth was found to be one. The needs of the LPs for relevant instructions left unreceived from the EDs were related to considering the colour of the amniotic fluid, loosening the umbilical cord around the baby's neck, the afterbirth, the closing of the umbilical cord and preparing for excreta:

*Because it looks so messy when the water breaks, and of course, there's some blood there as well* (Participant 2, LP).

**Rules:** The childbirth activity of the LPs was guided by the risk assessment protocol, where deviations sometimes produced disturbances. The LPs identified the disturbance in following the instruction, which was caused by the lack of a common language. The LPs felt that the disturbances could be related to the person giving birth coming from a different culture, and better knowledge of the structure of circumcised female genitalia was needed from the ED.

Disturbances related to childbirth in a car emerged in one LP's interview. They were linked to excessive speed when driving to the hospital, lack of space or the unsafety of driving immediately after the childbirth event:

*My speed was 220 kph, I had a phone on my ear, one hand on the wheel and the other hand in a way that the person giving birth could sink her nails in it* (Participant 3, LP).

In two LPs' interviews, the disturbance had been caused by the ED, who had interpreted the voluntary labour support person (doula) to be a professional in handling childbirth rather than an LP. This misinterpretation resulted in the cooperation situation being broken just before the moment of birth.

### Characteristics of and obstacles to negotiated knotworking

**Script innovations requiring midwifery competence.** In the EDs' stories (WSED and UWSED combined, n=31), the challenge level of an unplanned childbirth situation was increased by the difficulty of forming a situational picture over the phone:

*Different words and views pose challenges during calls whenever the dispatcher works with the layperson acting as their eyes. I try to handle these changing situations using different wordings and sentences so that the dispatcher can "see" the situation as it really is.* (Participant 29, UWSED).

Several development needs for a new kind of tool, video consultation, facilitating the formation of a situational picture, could be identified from the MEBS data. In the EDs' stories, the video connection was seen to extend the possibilities of

instructing LPs and achieving a positive outcome in future OHDs. These situations identified to support the use of the ED's sense of sight during labour included, for example, the visual examination of the presenting part to assess the progress of labour, the examination of the childbirth position, guidance of releasing the umbilical cord from around the neck, and the assessment of the condition of the newborn or the amount of lochia. In EDs' stories, a video connection would help prevent disturbances, especially in the absence of a common language or when the LP's attitude challenges the script. Using the sense of sight was considered necessary when the sounds of childbirth or traffic noise hindered verbal communication. The LP interview data could also identify innovation attempts to develop video consultations.

In the EDs' stories, instructing the LP in the OHD was considered particularly demanding because the training received for the task of childbirth had been brief, their expertise undeveloped, and the routine missing. However, most of the LPs interviewed relied on the ED's expertise in handling childbirth, even though they did not always know whether it existed. They felt that the ED had acted well in the childbirth situation and appreciated the ED's expertise in perceiving the overall picture – especially on the way to the hospital – and in alerting additional help. On the other hand, the EDs hoped that midwives familiar with genuine 791 emergency call recordings (791 is the code for a childbirth task) would provide additional and supplemental training for future childbirth tasks. The respondents wanted help from experienced midwives to update the risk assessment instructions.

It is significant that in the LPs' interviews, the midwife of the maternity hospital had, without exception, been called before the emergency call, sometimes several times, which describes the LPs' wish to primarily deal with a midwife in a situation related to childbirth:

> *When we left home, I called the maternity hospital and said we would be coming in pretty fast. Then I called them again before calling the emergency response centre* (Participant 3, LP).

However, according to the LP interviews, the telephone connection to the midwife had not been open in any of the cases at the moment of birth, either after the maternity hospital had placed the call on hold or because the midwife had directed the LP to call the emergency response centre after childbirth is noted to take place soon. Although the midwife of the maternity hospital had sometimes called the LP back after the childbirth, after the emergency call had ended, most LPs hoped that the ED would have involved the midwife in the call to support and instruct them in handling the OHD. Alternatively, they wished that the ED had been a midwife by training:

> *Nobody consulted the midwife at any point. The fact that you have been with women giving birth, you can immediately see what might happen. That there is someone who understands and knows it would be extremely important.* (Participant 2, LP).

Innovation attempts to develop video consultations, including the presence of both a midwife and the ED, could also be identified from the LP interview data, where both would simultaneously instruct and support the handling of the OHD.

**Emotion work.** Slightly more than half of the EDs and all LPs recognised that emotions were fundamentally related to the handling of rapidly progressing delivery tasks:

> *Childbirth is a time-restricted, sort of unstopping process in which big emotions are present* (Participant 18, UWSED).

In both EDs' stories and LPs' interviews, it had almost always been the LP that called the general emergency number 112. When childbirth was noted to take place before a health care professional could reach the site, the ED had to take into account the strong emotional charge connected to the situation and calm the LP down. In their stories, the EDs typically interpreted the emotional state of the LP as tense, nervous, anxious or panicked, which corresponded to the experiences

of most of the LPs interviewed. In particular, the fear of the LPs was increased by the fact that the person giving birth was someone close to them.

In the EDs' stories, the LP had to be calmed down so that they could listen and answer questions in the risk assessment protocol or receive instructions. Measures aimed at calming down the LP were motivating, encouraging, inspiring, and strengthening faith in a successful end result, which were highlighted especially if the person giving birth's condition deteriorated. Most LPs felt that the ED had succeeded in calming them down.

The words most frequently used in WSED stories were "informant" and "tells", which reflects the importance of calming down the LP for the query protocol to succeed. In these stories, the LP calming down often led to maintaining cooperation and functional capacity, the significance of which the EDs recognised as a prerequisite for achieving a successful result. All interviewed LPs felt that their functional capacity had remained – sometimes it had even improved in the childbirth event involving strong emotional stress:

> *My functional capacity went to the extreme, like the best possible condition: I will rock this to the finish and will not freeze* (Participant 3, LP).

The LP calming down was needed in order for the LP to reassure the person giving birth, who was often described by EDs in their stories as in their childbirth bubble, incapable of communicating or cooperating, distressed, panicked or screaming in terror. Many LPs felt that the person giving birth being reassured was achieved without using words, especially because the person was close to them. After the calming down, childbirth also often progressed rapidly.

In their stories, the EDs described being typically tense or even slightly panicked when receiving the childbirth task. However, this was not reflected in the interviews of most LPs, who rather appreciated the ED's calmness. The EDs had received mental support from other emergency dispatchers in the room who had observed the call.

In both the EDs' stories and the LPs' interviews, after childbirth, the LPs' emotional state typically changed to great relief. Despite this, almost all interviewed LPs felt that they needed debriefing.

The emotional state of the persons giving birth was described as having changed to relief or crying from happiness after the birth. However, the EDs' own tension in the stories had typically only been released when the baby's crying was heard:

> *At the same time, the situation is peaceful, but there is a small panic inside me, which, of course, I will not reveal to the layperson. Simultaneously, I'm going through the scenarios that could occur after the birth. What if the mother starts bleeding severely, or the child doesn't start crying.* (Participant 5, WSED).

In the WSED stories, the EDs' own tension when the mother and the newborn were physically well had typically turned into a great joy, which was often shared with other personnel in the emergency response centre who had listened to the call. The ED often thanked the participants, remembered to congratulate the person giving birth and the LP, and enquired about the baby's gender. However, even a successful childbirth event had often left both the ED and the LP thinking, both of whom felt small in the face of the miracle of birth. On the other hand, UWSED stories typically highlighted the ED's feelings of inadequacy and powerlessness, even quiet terror.

The findings of RQ1 and RQ2 are summarised in Fig 2, displaying activity systems in interaction and preliminarily identifying the themes of knotworking that shape and challenge their collaboration.

## Discussion

This study aimed to describe and analyse the perspectives of EDs and LPs as the ED instructs the LP in the handling of an OHD. In the activity theoretical framework, the ED and the LP are connected to each other through the mutually constructed object of activity [24]. In this study, an interesting result was that the object of the ED's activity was the physical

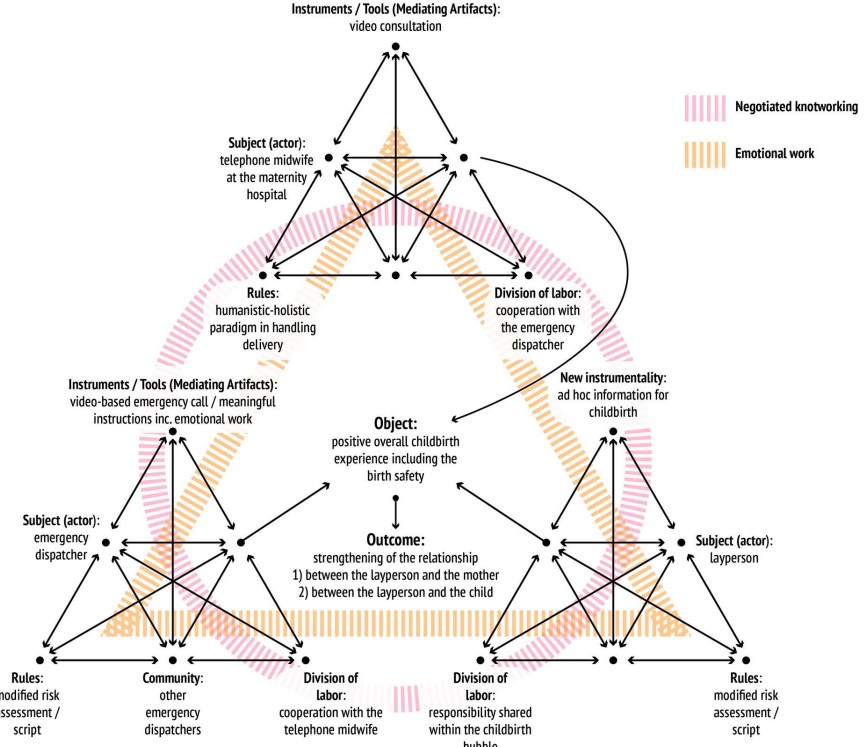

**Fig 2. Emergency dispatcher's expertise as negotiated knotworking.**

wellbeing of the person giving birth and the newborn, while the LP's object was the childbirth experience, including the aforementioned and shared with the person giving birth. There was little time to negotiate the content and meaning of the object [9].

In this study, the role of the mediating tool between the ED and the LP, the emergency call, was different for the parties. In Finland, the national risk assessment guidelines on the handling of health care tasks at the emergency response centre, the formal script of cooperation and the artifact regulating the course of interaction can be activity theoretically understood as a tool that has become a rule, i.e., a step-by-step guideline on how events should proceed from the starting point to the endpoint [24]. Disturbances and innovations in childbirth activity, observed by both EDs and LPs, can be interpreted as deviations from the script and expressions of developmental contradictions in activity systems, the resolution of which leads to qualitative changes in activity systems and the development of the expertise of maternity care agents [9].

The analysis of the material-instrumental and temporal dimensions of the discursive characteristics of the negotiated knotworking identified that the formal script of childbirth did not serve the negotiated knotworking [25]. The script defines the questions posed by the ED to the LP and the instructions given, highlighting its benefits, especially when the expertise in childbirth care is undeveloped. On the other hand, the risk assessment protocol may unnecessarily restrict the instruction provided by a childbirth professional who may function as the ED. The risk assessment protocol can be interpreted to represent the technocratic paradigm in managing childbirth and an adaptively quantifying perspective of biomedical authority, even if the qualitative knowledge and experiences of the LP are in question. Even if the risk assessment protocol would structure the interaction in an unplanned childbirth event, in this study, disturbances related to, for example, relevant instructions or giving birth in a car could be identified in the script of childbirth activity. Considering these disturbances increases the chances of constructing a shared object of activity, a positive overall childbirth experience.

The participants in an emergency call are people, meaning a significant amount of human activity is involved. Involving experienced midwives in developing the OHD risk assessment protocol could shape both the ED's and LP's rules guiding childbirth activity towards a humanistic-holistic paradigm [39].

In this study, the analysis of the ethical dimension discovered among the discursive characteristics of negotiated knot-working highlighted the significant study result that emotions [see 40] were fundamentally connected to the emergency call about an unplanned childbirth event. Apart from a tool in childbirth activity, emotion work [see 41] is a significant part of knot-like cooperation in order to both expand the material-instrumental and temporal dimensions in a time-restricted and unstoppable childbirth event. The emotional dimension does not explicitly emerge from applying the model of activity system, but the subjects brought up the strong emotional charge of the unplanned childbirth event for both the ED and the LP. In the ED's mentally stressful work, a strong emotional aspect was present in the object-orientation, in which case emotion work [42] may evolve into the employee's way of dealing with callers' fits of emotion. The aim of the emotion work in the emergency call is to keep the LP calm and act appropriately in a stressful situation. The emotional support helps the LP obtain the confidence to act with controlled determination in the OHD [13]. In emergency calls related to childbirth, which are content-wise heavy, the LP's emotional outbursts are handled, and the ED's own emotional reactions are suppressed. In fact, the emotional aspect should be better taken into account not only when developing the risk assessment protocol but also in order to assess the need for debriefing [see 43] or occupational counselling, especially after an unplanned childbirth event that has ended negatively.

This study extends the discussion on in-hospital deliveries and planned home deliveries [e.g., 44] towards OHDs. Midwives are not officially part of the Finnish emergency care service chain [2]. However, in an ideal situation, a specialist in delivery, a midwife, would instruct the LP in handling an OHD [45,46]. In this study, the analysis of the socio-spatial dimension of the discursive characteristics of negotiated knotworking identified the need for midwifery competence in unplanned childbirth events when the subjects discussed how to involve midwifery in knotworking. In an unexpected childbirth event, the LPs primarily aimed to deal with the midwife in the delivery room and made innovative attempts during the emergency call to obtain guidance from the midwife. However, the LP could not call the midwife again while the emergency call was in progress. Although Swedish-speaking emergency calls can be transferred to a language-proficient ED, emergency calls related to childbirth are not specifically transferred to, for example, an ED trained as a midwife or the telephone midwife of a delivery room. The curriculum for the Emergency Response Centre Operator degree has been reformed [47], but in this study, the EDs wanted there to be additional and supplemental training related to the instruction of handling childbirth provided by a midwife.

This study revealed that creating a situational picture related to an unplanned childbirth event over the phone is challenging from the perspective of EDs. For example, disturbances in the script caused by the lack of a common language or cultural differences [see 2] were brought up, for the solution of which the development of video consultation related to OHD would be justified. Even if the single emergency response centre authority model used in Finland approached expertise as a performance describing individual characteristics [see 48], in situations requiring highly specialised expertise, such as diving accidents, after an emergency call and sending help, it is recommended to contact, e.g., a specialist in diving medicine [see 49]. In emergency response centre operations, group calls are already utilised, for example, in interpreting services. Even if the internationally fairly unique model of one emergency response centre authority used in Finland would save time, an emergency call shared with the delivery room telephone midwife, which respects negotiated knotworking, and a video-based emergency call to instruct the LP in an unplanned childbirth situation, is a desirable option. It would combine dispatching additional assistance with the highest specialised expertise in instructing the handling of childbirth. The possibility of a midwife instructing an LP in the handling of childbirth remotely has been investigated in a Spanish study examining the possibilities offered by telemedicine and smart glasses in a simulated OHD. The video connection to the midwife significantly improved the chances of the LPs succeeding in handling an OHD [3]. The development work based on dialogue and mutual learning between agents may result in a self-assured subject and agent that crosses the boundaries of organisations and has a shared object of joint activity, a negotiated delivery cooperation knot.

Negotiated knotworking is the interactional core of co-configuration. Co-configuration as a special new form of production is related to interactional quality and is based on a customer-smart service that adapts to the user's activity and is modified cooperatively in a dialogue between service providers and customers [9,50]. In Finland, more than 2 million citizens use the 112 mobile application that transmits location information to the emergency response centre. Based on the results of this study, it was possible to identify the possibility of co-configuration in the multi-organisational field of OHD. As a long-life customer-smart IT solution adapting to the user's activity and requiring continuous redefinition, a new kind of dialogic informational tool containing video consultation could be developed, for example, a 791 delivery application, 791 is the code for a childbirth task, that corresponds to the already utilised diving accident application [see 49] to respond to OHDs.

## Methodological considerations

By examining an unplanned childbirth event from the perspective of more than one subject, it was possible to produce material suited to the question-setting of this study. The method of empathy-based stories (MEBS) was suitable for examining the perspectives of professionals [see 30]. The goal in recruiting EDs was to reach the entire target group when recruiting EDs, and they had the opportunity to win a spa gift card. However, the response rate remained low (>10%). Previous studies have found it challenging to get Finnish EDs to write stories. The collected MEBS data was considered sufficient content-wise, and the material was saturated [see 30]. However, the division of labour and outcome (RQ1) could not be analysed from the MEBS data.

The semi-structured interview was a way of giving voice to LPs when examining experiences that were meaningful to the target group. Both discretionary and snowball sampling were used to recruit the informants.

The analysis of the study data was performed thoroughly, using appropriate methodological solutions, meaning the data produced by this qualitative study can be considered reliable. As for reflexivity, the first author was aware of his own role as a researcher. The credibility of the results could be assessed by the results corresponding to the perspectives of the informants on the phenomenon being studied and they can be transferred to other similar situations. To achieve verifiability, the research process was documented so that other researchers would be able to follow it.

This study was conducted by a multidisciplinary research group. The first author is an experienced midwife and senior lecturer in maternity care with two Master's degrees and ongoing doctoral studies in education. He is a board member of the Finnish Association of Midwifery Sciences and a member of the Finnish Network for Birth and Childbearing Research (BIRRES). The second author is a professor of adult education at Tampere University, Finland. Her research focuses on learning in work-life networks applying the cultural-historical activity theory framework. The last author is a principal lecturer and adjunct professor with extensive experience in qualitative research. Her background is in health sciences, pedagogy, and disaster medicine. Her research interests have focused especially on the occupational wellbeing aspects of emergency medical service personnel and the functioning of the emergency medical service system. Multidisciplinarity, academic diversity and the affiliations of universities of sciences and universities of applied sciences strengthened the reliability of the study by offering not only expertise in educational and health sciences but also transversal theoretical-conceptual, theoretical-methodological and professionally oriented expertise in the phenomenon studied.

## Conclusions

The contribution of this study was to develop collaborative and transformative expertise as well as cooperation in the maternity care service system in responding to high-risk OHDs handled by LPs instructed by EDs. The results of this study will directly benefit not only the activity systems related to OHDs and their subjects but also society in the long term, for example, from the perspective of evening equality differences between different delivery models. However, the material limitation of this study was the lack of authentic 791 emergency call recordings related to OHDs, which is why their discourse analysis, especially from the perspectives of negotiated knotworking and emotion work, would be necessary in the

near future. In order to extend the object, solutions should be used to seek a new dialogic activity concept and a recommendation-based operational model. This draws attention to the value-based development of not only interaction and the expertise of professionals but also of the structured maternity care service system.

## Supporting information

**S1 Appendix.** Emergency dispatchers' (ED) frame stories.
(DOCX)

**S2 Appendix.** Semi-structured interview framework (LP).
(DOCX)

## Acknowledgments

We wish to thank the emergency dispatchers and laypersons who donated their time and provided their honest perspectives towards this research. We would like to thank Olli Lehtonen, Associate Professor from the University of Eastern Finland for text mining.

## Author contributions

**Conceptualization:** Jussi Hänninen, Hanna Toiviainen, Hilla Nordquist.

**Investigation:** Jussi Hänninen.

**Methodology:** Jussi Hänninen, Hanna Toiviainen, Hilla Nordquist.

**Supervision:** Hanna Toiviainen, Hilla Nordquist.

**Visualization:** Jussi Hänninen.

**Writing – original draft:** Jussi Hänninen.

**Writing – review & editing:** Hanna Toiviainen, Hilla Nordquist.

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
