## [Decision Letter · Decision Letter 0]

15 May 2025

PONE-D-25-15997Emergency dispatchers as instructors of laypersons in unplanned out-of-hospital deliveries - interdisciplinary qualitative studyPLOS ONE

Dear Dr. Hänninen,

Thank you for submitting your manuscript to PLOS ONE. After careful consideration, we feel that it has merit but does not fully meet PLOS ONE’s publication criteria as it currently stands. Therefore, we invite you to submit a revised version of the manuscript that addresses the points raised during the review process. The article needs some clarification / modifications in the manuscript as pointed out by two reviewers. 

We look forward to receiving your revised manuscript.

Kind regards,

Aamir Ijaz, MD, FCPS, FRCP, MCPS-HPE

Academic Editor

PLOS ONE

 [Publication and Production Services of the South-Eastern Finland University of Applied Sciences covers the Article Processing Charge].

3. In the online submission form, you indicated that [Availability of data and materials

The datasets generated and analyzed during the current study are not publicly available because they contain participant-identifying information, and are sensitive.].

4. Please include captions for your Supporting Information files at the end of your manuscript, and update any in-text citations to match accordingly. Please see our Supporting Information guidelines for more information: http://journals.plos.org/plosone/s/supporting-information .

Additional Editor Comments (if provided):

Reviewers' comments:

Reviewer's Responses to Questions

**Comments to the Author**

1. Is the manuscript technically sound, and do the data support the conclusions?

Reviewer #1: Yes

Reviewer #2: Yes

2. Has the statistical analysis been performed appropriately and rigorously? 

Reviewer #1: Yes

Reviewer #2: Yes

3. Have the authors made all data underlying the findings in their manuscript fully available?

Reviewer #1: Yes

Reviewer #2: No

4. Is the manuscript presented in an intelligible fashion and written in standard English?

Reviewer #1: Yes

Reviewer #2: Yes

5. Review Comments to the Author

Reviewer #1: Thanks for inviting me to review this interesting, but thought-provoking article. Although the outcome appears relatively subjective, this article is very unique and offers a different perspective on the issue of out-of-hospital delivery by laypeople in a developed clime. Apart from the fact that the title is not concise and catchy, the article is not only well-written, but will make an interesting read. I would recommend this article for publication.

Reviewer #2: The article is nicely written in context of how system works in Finland and will be useful for other countries using similar systems. However few minor statements need clarification.

1. In result section ,on line number 292 it is written that four out of five though total number of respondents were 31. This needs clarification

2.Line 323 placing the severely bleeding mother in recovery position is not clear and needs a bit of more explanation

6. PLOS authors have the option to publish the peer review history of their article (what does this mean? ). If published, this will include your full peer review and any attached files.

**Do you want your identity to be public for this peer review?** For information about this choice, including consent withdrawal, please see our Privacy Policy .

Reviewer #1: **Yes: ** Ekwuazi Kingsley Emeka

Reviewer #2: No

---

## [Author Response · Author response to Decision Letter 1]

17 Jun 2025

Dear Academic Editor of PLOS One,

Thank you for the opportunity to revise our manuscript “Emergency dispatchers as instructors of laypersons in unplanned out-of-hospital deliveries - interdisciplinary qualitative study”. We have carefully clarified the points raised by the Reviewers. We have marked the changes in the manuscript with yellow highlighting.

Regarding the funding of this research, the authors received no specific funding or salary for this work. Development Services unit of South-Eastern Finland University of Applied Sciences will pay the Article Processing Charge. The funder had no role in study design, data collection and analysis, decision to publish or preparation of the manuscript.

Regarding the public availability of the data, this study received a favorable opinion (Statement 127/2023, attachment) from the Ethics Committee of the Tampere Region during the advance evaluation on September 5, 2023. The ethical approval for this research did not include the public publication of the data. Therefore, making the data publicly available would breach compliance with the protocol approved by the research ethics board (PLOS ONE Data Availability Policy). It is not possible to make the data publicly available for ethical reasons, as it contains sensitive, health-related information that could compromise the participants’ privacy (ALLEA 2023; GDPR; TENK 2019). Health-related information (e.g., childbirth) constitutes special categories of personal data and must be handled with particular care, as it requires specific protection (GDPR). Other researchers can find more information about the Ethics Committee of the Tampere Region here: https://www.tuni.fi/en/research/responsible-science-and-research/research-integrity/ethics-committee-of-the-tampere-region and they can request access to the research data (available up-on request) via email:

Secretary: Senior specialist Heikki Eilo heikki.eilo@tuni.fi or researchdata@tuni.fi

Research authorization

The Emergency Response Centre Administration granted the research permit regarding the method of empathy-based stories (MEBS) on December 8, 2023 (HAK-2316452, attachment). The issued research permit did not include permission for the public publication of the research data. Although the MEBS data has been pseudonymized, a risk still exists of identifying individual participants or of sensitive, confidential health-related information being accessed by unauthorized parties.

The informed consent (ALLEA 2023; TENK 2019) provided by the emergency dispatchers for participation in this study did not include permission for the public publication of the research data, because they also refer to third parties (lay assistants and birthing persons). Health-related information (e.g., childbirth) constitutes a special category of personal data and must be handled with particular care, as it requires specific protection (GDPR). Other researchers can request access to the data from the Emergency Response Centre Agency https://112.fi/en/frontpage via email to:

Head of Project Management Office: Mrs. Ullamaija Nenonen ullamaija.nenonen@112.fi or pro-jektitoimisto@112.fi

The informed consent (ALLEA 2023; TENK 2019) provided by the lay assistants for participation in this study did not include permission for the public publication of the research data. Although the thematic interview data have been pseudonymized, there remains a risk of identifying individual participants or of sensitive, confidential health-related information being accessed by unauthorized parties. When obtaining informed consent from the lay assistants, permission for public data sharing was not requested, as this would not comply with the European GDPR, to which we must adhere. Opening the data retrospectively without informing the participants would not align with the good scientific practice as defined by the Finnish National Board on Research Integrity (TENK).

First authorship agreement

As the corresponding author and custodian of the data, I am committed to doing everything possible to ensure that other researchers are granted access to both MEBS and thematic interview data (available upon request) when necessary.

According to the research ethics and data policy of Tampere University, the research data can only be held by the researcher if they are not employed by the university. Since I am not in an employment relationship with Tampere University, I am the data controller for the research, not the university.

Personally, as a researcher, I support the PLOS ONE Data Availability Policy and the mission of Open Science. However, I am bound by ethical guidelines (ALLEA 2023; TENK 2019) and legislation (GDPR). The principle ‘as open as possible, as closed as necessary’ applies in this context (PLOS ONE Data Availability policy).

email: jussi.hanninen@xamk.fi

References

European Federation of Academies of Sciences and Humanities ALLEA (2023). https://allea.org/code-of-conduct/

Finnish National Board on Research Integrity TENK (2019). The Ethical Principles of Research with Human Participants and Ethical Review in the Human Sciences in Finland. Finnish National Board on Research Integrity TENK Guidelines 2019. https://tenk.fi/sites/default/files/2021-01/Ethical_review_in_human_sciences_2020.pdf

General Data Protection Regulation GDPR 9§. https://eur-lex.europa.eu/legal-content/FI/TXT/?uri=uriserv:OJ.L_.2016.119.01.0001.01.ENG&toc=OJ:L:2016:119:TOC

PLOS ONE Data Availability Policy. https://journals.plos.org/plosone/s/data-availability

We hope that our response adequately addresses all the points raised, and we look forward to hearing from you.

Sincerely,

Mr. Jussi Hänninen, Prof. Hanna Toiviainen, Dr. Hilla Nordquist

Reviewer One:

We truly appreciate the time you took to review our article and your encouraging feedback. Thank you.

Reviewer Two:

Thank you for taking the time to read our article and for your positive feedback.

On line 292, we have clarified that this is approximately 80% rather than 4 out of 5 participants. We hope this phrasing is clearer for the reader.

On line 323, we agree the phrasing about the “severely bleeding mother” was somewhat clumsy and have rewritten it as follows: “Relevant instructions related to handling the delivery included, for example, uncovering the lower body and observing the presenting part in the birth canal, instructing the person giving birth to push during a contraction, and if the mother experiences severe bleeding, she is placed into the recovery position.”

---

## [Editor Report · Decision Letter 1]

23 Jun 2025

Emergency dispatchers as instructors of laypersons in unplanned out-of-hospital deliveries - interdisciplinary qualitative study

PONE-D-25-15997R1

Dear Dr. Hänninen,

We’re pleased to inform you that your manuscript has been judged scientifically suitable for publication and will be formally accepted for publication once it meets all outstanding technical requirements.

Kind regards,

Aamir Ijaz, MD, FCPS, FRCP, MCPS-HPE

Academic Editor

PLOS ONE
---

## [Editor Report · Acceptance letter]

PONE-D-25-15997R1

PLOS ONE

Dear Dr. Hänninen,

I'm pleased to inform you that your manuscript has been deemed suitable for publication in PLOS ONE. Congratulations! Your manuscript is now being handed over to our production team.

Kind regards,

on behalf of

Professor Aamir Ijaz

Academic Editor

PLOS ONE